# Investigation of Mechanical Behaviors of Functionally Graded CNT-Reinforced Composite Plates

**DOI:** 10.3390/polym14132664

**Published:** 2022-06-29

**Authors:** Jin-Rae Cho, Young-Ju Ahn

**Affiliations:** 1Department of Naval Architecture and Ocean Engineering, Hongik University, Jochiwon, Sejong 30016, Korea; 2Department of Mechanical and Design Engineering, Hongik University, Jochiwon, Sejong 30016, Korea; yjahn70@hongik.ac.kr

**Keywords:** CNT-reinforced, functionally graded, composite plate, bending, free vibration, buckling load

## Abstract

In this paper, the mechanical behavior of a functionally graded carbon nanotube-reinforced composite (FG-CNTRC) plate is numerically investigated. According to the concept of a hierarchical model, the displacement is decomposed into the in-field functions and the assumed thickness-wise monomial. The former is defined on the plate midsurface and is approximated by the 2-D meshfree natural element method (NEM). The FG-CNTRC plate is modeled as a homogenized orthotropic body, and its effective elastic properties are determined by referring to MD simulation and the linear rule of mixtures. Regarding the thickness-wise distribution of CNTs, one uniform and three functionally gradient distributions are taken. Through comparative numerical experiments, the reliability of the numerical method is justified with the maximum relative difference of 6.12%. The effects of the volume fraction and vertical distribution of CNTs, the plate width-thickness and aspect ratios, and the boundary conditions on the bending, free vibration, and buckling behaviors of FG-CNTRC plates are examined. It is found that the mechanical behavior of FG-CNTRC plates is significantly dependent of these major parameters.

## 1. Introduction

Carbon nanotubes (CNTs) have been spotlighted as an innovative material for the 21st century due to their outstanding thermo–mechanical properties [1]. They are fabricated by rolling graphene sheets into the form of a cylinder and are divided into single- and multi-walled carbon nanotubes according to the number of graphene sheets. The excellent properties not only promise CNTs as next-generation multi-functional reinforcements for polymer composites [2] but also remarkably extend the conventional polymer composites to a variety of engineering fields. As a representative application, carbon nanotube-reinforced composites (CNTRCs) were recently introduced, and extensive studies have focused on their fabrication methodology and thermo–mechanical behavior [3]. A simple technique for fabricating the aligned arrays of CNTs was presented by Ajayan et al. [4] by utilizing cutting thin slices. The elastic properties of CNTRC were predicted by Hu et al. [5] through structural deformation analyses for various loading conditions and were also evaluated by Han and Elliott [6] through molecular dynamics (MD) simulation.

However, the success in applying CNTRCs to engineering applications relies on the in-depth investigation of their thermo–mechanical behaviors because CNTRCs can have various forms of structural elements under various loading and boundary conditions. Wuite and Adali [7] presented the mechanical responses of a CNTRC beam that were predicted by multiscale simulation using laminated beam theory. Formica et al. [8] presented an equivalent continuum theory to investigate the vibration behavior of CNTRCs, and Arani et al. [9] investigated the buckling behavior of laminated CNTRC plates by combining analytical and FE methods. Wang and Shen [10] applied the higher-order shear deformation theory (SDT) to the non-linear vibration analysis of a CNTRC plate resting on an elastic foundation. Through these early studies, it was found that the target performance of CNTRCs reached a maximum when CNTs were parallel with the direction of the external load. Therefore, the conventional uniform dispersion of CNTs within a polymer matrix encounters a limitation in the improvement of the thermo–mechanical performance [10,11].

The conventional fabrication method for nano-composites allowed only uniform or random CNT dispersion, which naturally provided CNRTCs with uniform elastic properties in space. However, this restriction in producing the spatial material-property distribution can be resolved if the concept of functionally graded material (FGM) [12] is adopted. In FGMs, the volume fractions of the base constituents through the thickness can be artificially designed such that the target performance reaches its maximum. Smaranda et al. [13] prepared composites under the free membrane form and reported the structural and conductive properties of these new composite materials. Motivated by the concept of FGMs, Shen [14] and Ke et al. [15] presented a purposeful CNT distribution with CNTRCs through the thickness for the sake of bending deformation suppression and vibration control. Among the functionally graded CNT distributions, the three most representatives are FG-V, FG-O, and FG-X. These CNTRCs with functional CNT distributions are called FG-CNTRCs, and these became a hot research subject for many subsequent investigators [16,17,18,19,20].

More recently, Civalek and Avcar [21] analyzed the free vibration and buckling behaviors of FG-CNTRC-laminated non-rectangular plates using a four-node straight-sided transformation method. Arefi et al. [22] presented the size-dependent deflection analysis of FG-CNTRC micro-plates by applying the third-order shear deformation theory to the principle of virtual work. Gopalan et al. [23] attempted to experimentally and numerically investigate the dynamic characteristics of flax/bio epoxy functionally laminated composite plates and performed an optimization study using the response surface method (RSM). Cheshmeh et al. [24] investigated the buckling and free vibration of FG-CNTRC rectangular plates using the higher-order shear deformation theory. Alazwari et al. [25] investigated the effect of thickness stretching on the free vibration, bending, and buckling behavior of FG-CNT-reinforced composite nanoplates using a four-unknown quasi-3D higher-order shear deformation theory.

However, the mechanical behavior of FG-CNTRCs still needs exploration, particularly the effects of major parameters on the bending, free vibration, and buckling behaviors. Moreover, the application of NEM, the latest introduced meshfree method, to FG-CNTRC plates has rarely been presented. In this context, this study intends to establish an NEM-based 2-D numerical method for parametrically and profoundly examining the mechanical behavior of 3-D FG-CNTRC plates. The displacement is expressed as a product of the in-plane vector functions and the thickness monomials, according to the concept of the hierarchical model [26]. The former is defined based on the midsurface of the plate and is solved by the natural element method [27], while the latter is assumed a priori. The FG-CNTRC plate is viewed as an orthotropic body, and its effective mechanical properties are determined by referring to the MD simulation and the rule of mixtures. Through the numerical experiments, the validity of the present method is justified. The bending deformation, free-vibration, and buckling behaviors of FG-CNTRC plates are investigated as the major parameters.

## 2. Modeling of FG-CNTRC Plates

Figure 1a depicts a rectangular polymer plate in which SWCNTs are inserted uniformly through the thickness, where CNTs are parallel to the x− direction and the dimensions of the plate are length a, width b, and thickness d. Figure 2b represents three different functionally graded (FG) CNT distributions: FG-V, FG-O, and FG-X. Since CNTRC plates can be viewed as a two-phase composite, their equivalent elastic properties can be estimated using the rules of mixtures or the Mori-Tanaka method [12]. In this study, a modified linear rule of mixtures in which the CNT efficiency parameters $\eta_{j}\left(j=1,2,3\right)$ are introduced is employed. Based on this method, the equivalent elastic and shear moduli of the FG-CNTRC plate are calculated as follows [13]:(1)$$E_{1}=\eta_{1}V_{CNT}E_{1}^{CNT}+V_{m}E^{m}$$
(2)$$\frac{\eta_{2}}{E_{2}}=\frac{V_{CNT}}{E_{2}^{CNT}}+\frac{V_{m}}{E^{m}}$$
(3)$$\frac{\eta_{2}}{G_{12}}=\frac{V_{CNT}}{G_{12}^{CNT}}+\frac{V_{m}}{G^{m}}$$

$V_{CNT}$ and $V_{m}=1-V_{CNT}$ indicate the volume fractions of CNTs and the polymer matrix, respectively. In Equations (1)–(3), the orthotropic elastic properties of CNT are labeled as CNT while those of the isotropic polymer matrix are denoted by m. The scale-dependence of the equivalent elastic properties of CNTRCs is reflected by the CNT efficiency parameters $\eta_{j}$ which were calculated by equating the equivalent elastic properties predicted by MD simulation with those obtained by the rule of mixtures [28].

The volume fraction $V_{CNT}$ of carbon nanotubes in the UD- and FG-CNTRC plates through the thickness is given by
(4)$$V_{CNT}\left(z\right)=\{\begin{array}{ll} V_{CNT}^{*}, & \mathrm{UD}\;\\ \left(1+2z/d\right)V_{CNT}^{*}, & \mathrm{FG}-V\;\\ 2\left(1-2\left|z\right|/d\right)V_{CNT}^{*} & \mathrm{FG}-O\\ 2\left(2\left|z\right|/d\right)V_{CNT}^{*} & \mathrm{FG}-X \end{array}$$
with
(5)$$\frac{1}{V_{CNT}^{*}}=1+\frac{\rho^{CNT}}{\rho^{m}}\left(\frac{1}{w_{CNT}}-1\right)$$

Here, $w_{CNT}$ denotes the mass fraction of CNTs within the CNTRC plate, and $\rho^{m}$ and $\rho^{CNT}$ are the densities of the matrix and CNT, respectively.

In a similar manner, the equivalent Poisson’s ratio $\nu_{12}$ and the equivalent density ρ of the CNTRC plate are calculated as follows:(6)$$\nu_{12}=V_{CNT}\nu_{12}^{CNT}+V_{m}\nu_{12}^{m}$$
(7)$$\rho =V_{CNT}\rho_{}^{CNT}+V_{m}\rho_{}^{m}$$

## 3. Analysis of Bending, Free-Vibration, and Buckling

The displacement $u={\left\{u_{x},u_{y},u_{z}\right\}}^{T}$ in the bending, free-vibration, and buckling of the FG-CNTRC plate is expressed by a (1,1,0)* hierarchical model [26], which is equivalent to the first-order SDT.
(8)$$u_{x}\left(x\right)=\Theta_{x}^{0}\left(x,y\right)+\Theta_{x}^{1}\left(x,y\right)\left(\frac{2z}{d}\right)=\sum_{m=0}^{1}\Theta_{x}^{m}\left(x,y\right){\left(\frac{2z}{d}\right)}^{m}$$
(9)$$u_{y}\left(x\right)=\Theta_{y}^{0}\left(x,y\right)+\Theta_{y}^{1}\left(x,y\right)\left(\frac{2z}{d}\right)=\sum_{m=0}^{1}\Theta_{y}^{m}\left(x,y\right){\left(\frac{2z}{d}\right)}^{m}$$
(10)$$u_{z}\left(x\right)=\Theta_{z}^{0}\left(x,y\right)=\sum_{m=0}^{0}\Theta_{z}^{m}\left(x,y\right){\left(\frac{2z}{d}\right)}^{m}$$

Here, $\Theta_{\alpha }^{m}\left(x,y\right)$ and ${\left(2z/d\right)}^{m}$ are 2-D in-plane vector functions and 1-D assumed thickness monomials, respectively. Figure 2a demonstrates a uniform NEM grid consisting of N grid points (nodes) and M Delaunay triangles. The NEM grid is generated on the plate midsurface ω (see Figure 1a). For each grid point, the Laplace interpolation (L/I) function $\varphi_{J}\left(x\right)$ illustrated in Figure 2b is assigned with the help of Delaunay triangulation and the Voronoi diagram [27,29,30]. With a finite number of L/I functions, both actual and virtual displacements $u^{h}$ and $v^{h}$ are approximated as
(11)$$u_{\alpha }^{h}\left(x\right)=\sum_{m=0}^{q_{\alpha }}\left(\sum_{J=1}^{N}U_{\alpha ,J}^{m}\varphi_{J}\left(x,y\right)\right)\cdot {\left(\frac{2z}{d}\right)}^{m}$$
(12)$$v_{\beta }^{h}\left(x\right)=\sum_{\ell =0}^{q_{\beta }}\left(\sum_{I=1}^{N}V_{\beta ,I}^{\ell }\varphi_{I}\left(x,y\right)\right)\cdot {\left(\frac{2z}{d}\right)}^{\ell }\;$$

Here, $\left(q_{x},q_{y},q_{z}\right)=\left(1,1,0\right)$ and $U_{\alpha ,J}^{m}$ are the nodal displacements corresponding to the thickness monomial ${\left(2z/d\right)}^{m}$ of $u_{\alpha }$ at node J.

Under the assumption of $q_{x}=q_{y}=q_{z}=q$ for the convenience of concise expression, the virtual strain vector $\varepsilon \left(v^{h}\right)$ and the actual stress vector $\sigma \left(u^{h}\right)$ are denoted as
(13)$$\varepsilon \left(v^{h}\right)=\sum_{\ell =0}^{q}\sum_{J=1}^{N}B_{J}^{\ell }U_{J}^{\ell }\cdot {\left(\frac{2z}{d}\right)}^{\ell }$$
(14)$$\sigma \left(u^{h}\right)=D\varepsilon \left(u^{h}\right)=\sum_{m=0}^{q}\sum_{J=1}^{N}DB_{J}^{m}U_{J}^{m}\cdot {\left(\frac{2z}{d}\right)}^{m}$$
with the (6×6) orthotropic elastic constant matrix D [31]. The partial differential operator $B_{J}^{\ell }$ is defined by
(15)$$B_{J}^{\ell }={\left[\begin{array}{cccccc} \varphi_{J,X} & 0 & 0 & \varphi_{J,y} & 0 & \ell \varphi_{J}/z\\ 0 & \varphi_{J,y} & 0 & \varphi_{J,x} & \ell \varphi_{J}/z & 0\\ 0 & 0 & \ell \varphi_{J}/z & 0 & \varphi_{J,y} & \varphi_{J,x} \end{array}\right]}^{T}$$
with $\varphi_{J,\alpha }=\partial \varphi_{j}/\partial \alpha$. Then, the principle of virtual work leads to the coupled simultaneous equations defined by
(16)KU=F
to compute the nodal displacements ${\left\{U\right\}}_{\beta ,J}^{m}={\left\{U_{x.J}^{m},U_{y,J}^{m},U_{z,J}^{m}\right\}}^{T}$.

In Equation (16), the stiffness matrix K and the load vector F are defined by
(17)$${\left[K\right]}_{\alpha \beta ,IJ}^{\ell m}=\int_{-d/2}^{d/2}\left[\int_{\omega }\left\{\left(B_{I}^{T}D_{1}B_{J}\right)+{\left(B_{I}^{T}D_{2}B_{J}\right)}_{RI}\right\}d\omega \right]\cdot {\left(\frac{2z}{d}\right)}^{\ell +m}dz$$
(18)$${\left\{F\right\}}_{\alpha ,I}^{\ell }=\int_{\partial \Omega_{N}}t_{\alpha }\varphi_{I}ds\cdot {\left(2z^{*}/d\right)}^{\ell }$$

The subscript *RI* in Equation (17) denotes the selectively reduced integration (SRI) [32,33] using one Gauss point to avoid shear locking for the bending-dominated thin structure. The numerical integration in Equation (17) is carried out by combining the thickness-wise analytical integration and the in-plane Gauss numerical integration. For the SRI integration, the elastic constant matrix D should be divided as follows:(19)$$D_{1}^{}=\left[\begin{array}{cccccc} C_{11} & C_{12} & C_{13} & 0 & 0 & 0\\ C_{12} & C_{22} & C_{23} & 0 & 0 & 0\\ C_{13} & C_{23} & C_{33} & 0 & 0 & 0\\ 0 & 0 & 0 & G_{12} & 0 & 0\\ 0 & 0 & 0 & 0 & 0 & 0\\ 0 & 0 & 0 & 0 & 0 & 0 \end{array}\right],\;\;\;\;\;D_{2}^{}=diag\left(0,0,0,0,G_{23}^{\prime },G_{31}^{\prime }\right)$$
where $G_{23}^{\prime }=G_{23}/\kappa$ and $G_{31}^{\prime }=G_{31}/\kappa$ with κ=6/5 being the modified shear correction factor.

For the free-vibration analysis of the FG-CNTRC plate, the harmonic response $u\left(x;t\right)=\bar{u}\left(x\right)\cdot e^{j\omega t}$ is considered, together with the mass matrix ${\left[M\right]}_{\alpha \beta ,IJ}^{\ell m}$ defined by
(20)$${\left[M\right]}_{\alpha \beta ,IJ}^{\ell m}=\int_{-d/2}^{d/2}\left[\int_{\omega }\rho \left(z\right)\left(\varphi_{I}I\right)\left(\varphi_{J}I\right)d\omega \right]\cdot {\left(\frac{2z}{d}\right)}^{\ell +m}dz$$
with the (3×3) identity matrix I. Then, the weak formulation of the eigenvalue problem provides the modal equations given by
(21)$$\left(K-\omega^{2}M\right)\bar{U}=0$$
for computing the natural frequencies ${\left\{\omega_{J}\right\}}_{J=1}^{N}$ and natural mode s ${\left\{{\bar{U}}_{J}\right\}}_{J=1}^{N}$.

Next, the following virtual work principle is considered to derive the discretized equations of the linear buckling problem:(22)$$\delta U-\delta W_{ex}=0$$

Here, the virtual strain energy δU and the virtual work $\delta W_{ex}$ are given as follows:(23)$$\delta U=\int_{-d/2}^{d/2}\int_{\omega }\delta \varepsilon^{T}\sigma d\omega dz$$
(24)$$\delta W_{ex}=\int_{\omega }\delta {\left\{\begin{array}{l} u_{z,x}\\ u_{z,y} \end{array}\right\}}_{z=0}^{T}\left[\begin{array}{cc} N_{x}^{0} & 0\\ 0 & N_{y}^{0} \end{array}\right]{\left\{\begin{array}{l} u_{z,x}\\ u_{z,y} \end{array}\right\}}_{z=0}d\omega$$

Substituting the previous Equations (11)–(14) into Equations (23) and (24) results in the eigenvalue equations given by
(25)$$\left(K-\lambda K_{G}\right)U=0$$
to calculate the buckling loads ${\left\{\lambda_{J}\right\}}_{J=1}^{N}$ and the associated buckling modes ${\left\{U_{J}\right\}}_{J=1}^{N}$. Here, the geometric stiffness matrix $K_{G}$ is defined by
(26)$${\left[K_{G}\right]}_{IJ}^{\ell m}=\int_{\omega }\left\{\varphi_{J,x}\varphi_{J,y}\right\}\left[\begin{array}{cc} \gamma_{1}\left(=1\right) & 0\\ 0 & \gamma_{2} \end{array}\right]\left\{\begin{array}{l} \varphi_{I,x}\\ \varphi_{I,y} \end{array}\right\}d\omega \cdot {\left(\frac{2z}{d}\right)}_{z=0}^{\ell +m}$$

Here, $\left(\gamma_{1},\gamma_{2}\right)=\left(1,0\right)$ indicates the uniaxial buckling while $\left(\gamma_{1},\gamma_{2}\right)=\left(1,1\right)$ denotes the biaxial buckling. The lowest eigenvalue $\lambda_{1}$ becomes the critical buckling load $N_{cr}$, and the in-plane loads $N_{x}^{0}$ and $N_{y}^{0}$ are $\gamma_{1}N_{cr}$ and $\gamma_{2}N_{cr}$, respectively.

## 4. Results and Discussion

Figure 3a shows a simply supported FG-CNTRC plate subject to a uniform distributed load $q_{0}$ equal to $1.0\times 10^{5}\;N/m^{2}$. The width a is 0.1 m and the thickness d is a set variable for the sake of parametric investigation. The matrix is PmPV [9] and its isotropic elastic properties are $E^{m}=2.1\;GPa$, $\nu^{m}=0.34$, and $\rho^{m}=1150\;kg/m^{3}$. CNTs have the orthotropic material properties given by $E_{1}^{CNT}=5.6466\;TPa,E_{2}^{CNT}=7.080\;TPa$, $v_{12}^{CNT}=0.175,$
$G_{12}^{CNT}=1.9445\;TPa$, and $\rho^{CNT}=1400\;kg/m^{3}$, respectively. It is assumed that $E_{3}^{CNT}=E_{2}^{CNT},$
$\nu_{23}^{CNT}=\nu_{31}^{CNT}=0$, and $G_{23}^{CNT}=G_{31}^{CNT}=G_{12}^{CNT}$.

The CNT efficiency parameter $\eta_{j}$ for the numerical simulations was chosen by referring to Zhu et al. [20] and is recorded in Table 1. Figure 3b shows a 21×21 uniform NEM grid generated on the midsurface of the plate. The stiffness and mass matrices and load vector were integrated by making use of the 2-D in-plane Gaussian integration using 7 points and the thickness-wise trapezoidal rule using 40 equal segments. The simply supported condition is implemented as follows: ${\bar{U}}_{z}^{0}=0$ for all sides of the midsurface, ${\bar{U}}_{x}^{0}={\bar{U}}_{x}^{1}=0$ for two sides ① and ③, and ${\bar{U}}_{y}^{0}={\bar{U}}_{y}^{1}=0$ for the other two sides ② and ④.

The bending analyses were carried out by changing the value of $V_{CNT}^{*}$, the width-thickness ratio a/d, and the CNT distributions. The calibrated central deflections $u_{z}^{c}/d$ are recorded in Table 2 for comparison with those of Zhu et al. [19]. First of all, one can see that the present method is in good agreement with Zhu et al., with the peak relative difference of 1.01% at FG-X for $V_{CNT}^{*}=0.11$ and a/d=10. In addition, the difference between the two methods becomes smaller and is proportional to a/d. This agrees with the fact that all the plate theories approach a certain limit theory (for example, the Kirchhoff theory for isotropic plates) as the thickness decreases [26].

Figure 4a comparatively represents the magnitude of $u_{z}^{c}/d$ for three different values of $V_{CNT}^{*}$, where the case without reinforcement with CNTs is included. First, it is apparent that the central deflection is significantly reduced by reinforcing only a small amount of CNTs (more than seven times at a/d=50 when compared with the non-reinforced plate). Second, the magnitude of $u_{z}^{c}/d$ becomes smaller in proportion to $V_{CNT}^{*}$. Figure 4b represents the effect of the vertical CNT distribution on the magnitude of $u_{z}^{c}/d$, where FG-X produces the lowest value while FG-O leads to the highest one. FG-V and UD show the second and third highest levels, respectively. This relative variation of $u_{z}^{c}/d$ can be understood from the fact that the bending stiffness is influenced by the vertical distribution pattern of CNTs. FG-X has the highest bending stiffness because CNTs are concentrated on the bottom and top sides, and vice versa for FG-O. This result informs us that the desired mechanical behavior of the FG-CNTRC plate can be obtained when the vertical CNT distribution is suitably designed. 

Figure 5a comparatively shows the thickness-wise distributions of calibrated normal stress ${\bar{\sigma }}_{xx}=\sigma_{xx}d^{2}/\left(q_{0}a^{2}\right)$ to the vertical CNT distribution pattern. The width-thickness ratio a/d and $V_{CNT}^{*}$ are 50 and 0.17, and the stresses were measured along the thickness at point O. UD shows a typical anti-symmetric linear variation, but three different FGs exhibit curved nonlinear variations. It is because the (1,1,0)* hierarchical model leads to liner axial stress distribution through the thickness of the homogeneous material. However, for the inhomogeneous material, the vertical distribution of the axial stress is dependent on the vertical distribution of CNTs. FG-O and FG-X produce anti-symmetric variation, but FG-V shows a leaning parabolic variation. This is caused by the thickness-wise distribution of CNTs, which characterizes the bending stiffness along with the thickness. Figure 5b comparatively shows the thickness-wise distributions of shear stress ${\bar{\tau }}_{zx}=\tau_{zx}d^{2}/\left(q_{0}a^{2}\right)$ at point O. As in the previous Figure 5b, FG-O represents the highest value and FG-X provides the lowest value owing to the difference in the bending stiffness among the four CNTRC plates. Differing from UD, FG-O, and FG-X, FG-V exhibits a leaning shear stress distribution owing to the non-symmetric vertical distribution of CNTs, as in ${\bar{\sigma }}_{xx}$ in Figure 5a. Therefore, this result confirms once again that the behavior of bending deformation and stress is strongly influenced by the thickness-wise distribution of the CNTs.

Next, the free-vibration behavior of the CNTRC plate was examined under the value of $V_{CNT}^{*}$, the width-thickness ratio a/d, and the CNT distribution pattern. A simply supported square FG-CNTRC plate shown in Figure 4a was taken, using the same material properties for the matrix and CNTs. However, the NEM grid density was changed from 21 × 21 to 41 × 41 in order to secure the modal analysis accuracy. A total of twenty-one modes were extracted from the Lanczos and Jacobi iterations. In Table 3, the six lowest calibrated natural frequencies $\bar{\omega }=\omega \left(a^{2}/d\right)\sqrt{\rho^{m}/E^{m}}$ are compared with those of Zhu et al. [20]. First, it can be observed that the natural frequencies of modes (2,1) and (2,2) are higher than those of modes (1,3) and (1,4), regardless of $V_{CNT}^{*}$ and the thickness-wise CNT distribution. This mode sequence of FG-CNTRC plates is not the same as that of isotropic plates [34], which is due to the difference in the elastic properties between the x- and y-axes. The elastic properties of the y-axis are smaller than those of the x-axis because CNTs are parallel to the x-direction, as depicted in Figure 1a. Second, the comparison of detailed natural frequencies between the two methods informs us that the peak relative difference equal to 6.07% occurs at $V_{CNT}^{*}=0.14$ of FG-V. Thus, the accuracy of the present method has been verified.

Figure 6a comparatively represents the variation in the calibrated fundamental frequencies (CFFs) ${\bar{\omega }}_{\left(1,1\right)}$ to the volume fraction $V_{CNT}^{*}$. It is found that CFFs increase in proportion to $V_{CNT}^{*}$ because the stiffness increase is superior to the mass increase. This explanation can be justified from the fact that the non-CNTRC plate (i.e., $V_{CNT}^{*}=0$) provides a much lower CFF. Figure 6b compares CFFs with respect to the thickness-wise CNT distributions. By referring to the previous Figure 5b for the calibrated central deflection, one can realize that the trend is completely reversed. That is, FG-X exhibits the highest CFF while FG-O provides the lowest one. UD and FG-V show the second and third highest levels, respectively. Thus, it has been justified once again that the vertical distribution of CNTs is important, and FG-X is the stiffest while FG-O is the weakest. The results of Figure 7a,b inform us that the vibration characteristic of the CNTRC plate is markedly influenced by the volume fraction $V_{CNT}^{*}$ and the thickness-wise CNT distribution.

Next, the critical buckling loads of the FG-CNTRC plate were calculated and compared with the analytical solutions of Chesmeh et al. [24], as given in Table 4. The plate was simply supported, its width–thickness ratio a/d was 100, and three different vertical distributions and volume fractions of CNTs were taken. The critical buckling loads were calibrated as ${\bar{N}}_{cr}=N_{cr}a^{2}/E_{m}d^{3}$. The midsurface of the plate was uniformly discretized by 15×15 grid points, and the lowest six buckling load parameters and buckling modes were solved by Lanczos transformation and Jacobi iteration. It was found that the calibrated buckling loads obtained by the present NEM were slightly higher than the analytical solutions by Chesmeh et al. [24]. This is consistent with the fact that the numerical approximation provides buckling loads that are higher than the analytical solutions. The maximum relative error occurred at the uniaxial for $V_{CNT}^{*}=0.17$ of FG-O, and its magnitude was 6.12%. Therefore, the accuracy of the present method has been justified once again.

Figure 7a compares the variations of ${\bar{N}}_{cr}$ to the width-thickness ratio a/d for various distributions of CNTs. First of all, one can see that the calibrated buckling loads increase and saturate as a/d becomes larger. Regarding the vertical CNT distribution, the highest and lowest levels occur at FG-X and FG-O and the second and third highest levels occur at UD and FG-V, respectively. This trend is almost the same as one of the previous free vibrations, and it is because the buckling load increases in proportion to the plate stiffness, which is influenced by the vertical CNT distribution. This trend is also observed in Figure 7b, which represents the influence of the CNT volume fraction $V_{CNT}^{*}$ on the calibrated buckling load ${\bar{N}}_{cr}$. It can be seen that ${\bar{N}}_{cr}$ uniformly increases in proportion to the value of $V_{CNT}^{*}$. Therefore, the buckling could be effectively suppressed by increasing $V_{CNT}^{*}$ and/or adopting FG-X.

Figure 8a represents the influence of boundary conditions on the variation of ${\bar{N}}_{cr}$ to the width-thickness ratio a/d, where S and C denote simply supported and clamped. A combination of four characteristics denotes a combination of boundary conditions that is enforced to sides ➀, ➁, ➂, and ➃ shown in Figure 3. The highest and lowest levels are seen at CCCC and SSSS, while SCSC and CSCS provide the curves close to CCCC and SSSS, respectively. It is because two sides (➁ and ➃) play an important role in the boundary condition for the FG-CNTRC plates in which CNTs are parallel to the x− direction, and these two sides are clamped at SCSC while simply supported at CSCS.

Figure 8b compares the variations in ${\bar{N}}_{cr}$ with the aspect ratio a/b for different boundary conditions, where the CNT distribution is UD and the width-thickness ratio a/d is 10. One can see that the buckling load increases and saturates as the aspect ratio a/b becomes larger because the plate becomes stiffer as the constrained plate becomes narrower with the side length a kept the same. The difference in ${\bar{N}}_{cr}$ between boundary conditions becomes insensitive as the aspect ratio increases. This is because the width of the two dominant sides ➁ and ➃ becomes shorter as the aspect ratio increases so that the type of boundary condition specified for these sides does not lead to a marked difference.

## 5. Conclusions

The bending, free-vibration, and buckling behaviors of FG-CNTRC plates were investigated by 2-D NEM. Based on the hierarchical model, the displacement was split into the 2-D in-plane vector functions and the assumed thickness monomials. The in-plane displacement part was solved by the 2-D natural element method. The numerical experiments were performed to validate the introduced method and to explore the bending, free-vibration, and buckling characteristics of the FG-CNTRC plates. The following observations were drawn from the numerical results:The accuracy of the present method is justified with the peak relative differences between the proposed method and the references, which are 1.01% for the central deflection, 6.07% for the natural frequencies, and 6.12% for the buckling loads.The central deflection is significantly reduced by introducing only a small amount of CNTs, and its reduction is proportional to $V_{CNT}^{*}$. The natural frequencies and buckling loads show a completely reverse trend.The (2,1) and (2,2) modes show higher frequencies than the (1,3) and (1,4) modes because CNTs are parallel to the x-direction. In other words, the stiffness in the x-direction is much higher than that in the y-direction.UD, FG-O, and FG-X produce anti-symmetric normal stress distributions and symmetric shear stress distributions; FG-V does not show such distribution but rather a leaning parabolic distribution.The buckling load is sensitive to the type of boundary condition, and it increases and saturates in proportion to the aspect ratio. The difference in the buckling load between boundary conditions becomes insensitive in proportion to the aspect ratio.The comparison of the magnitudes of the central deflection, fundamental frequency, and buckling load shows that the order of stiffness of the FG-CNTRC plates is FG-X > UD > FG-V > FG-O.

The current study was conducted from the macroscopic point of view. However, the microscopic factor such as the microstructural evolution would be important, and this represents a topic that deserves future work.

## Figures and Tables

**Figure 1 polymers-14-02664-f001:**
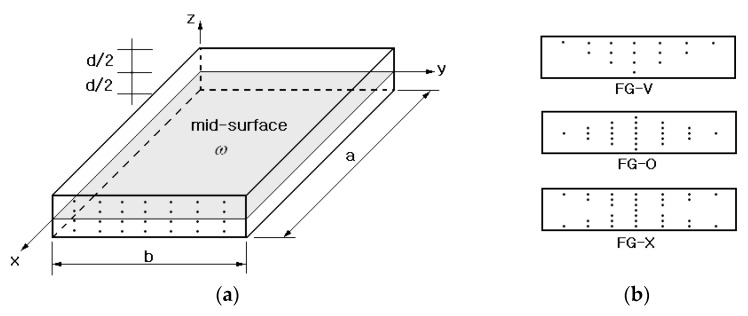
CNT-reinforced composite: (**a**) uniform distribution (UD); (**b**) functional gradient (FG).

**Figure 2 polymers-14-02664-f002:**
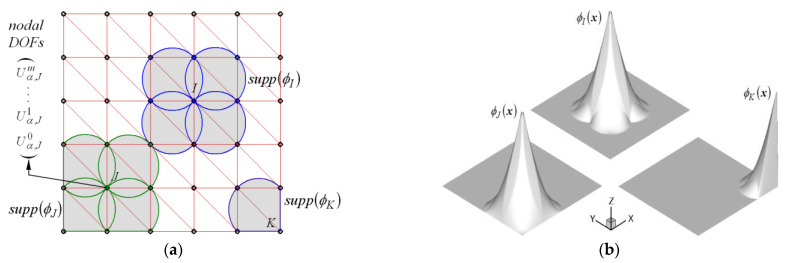
NEM: (**a**) uniform NEM grid; (**b**) L/I functions.

**Figure 3 polymers-14-02664-f003:**
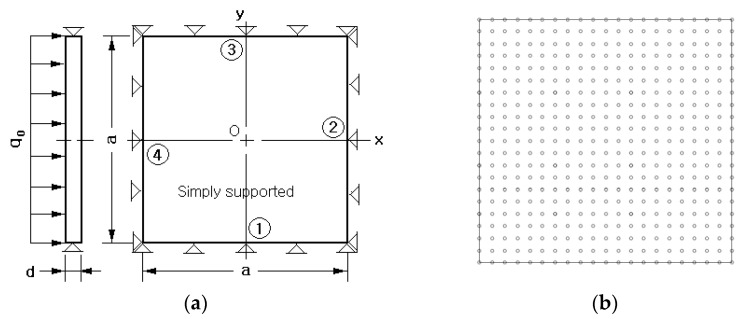
A square CNTRC plate under a uniformly distributed load $q_{0}$: (**a**) geometry and dimensions; (**b**) a 21×21 uniform NEM grid.

**Figure 4 polymers-14-02664-f004:**
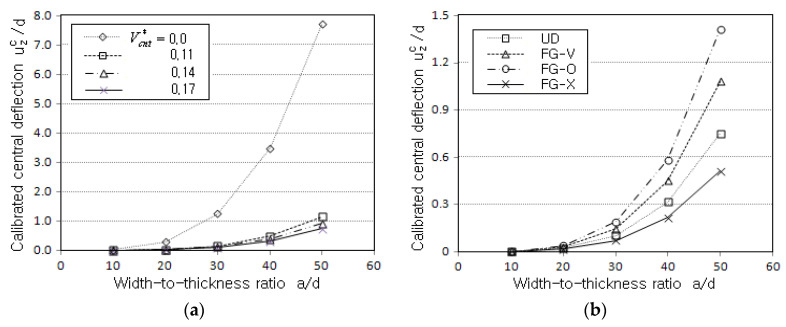
Comparison of calibrated central deflection $u_{z}^{c}/d$: (**a**) to $V_{CNT}^{*}$ (SSSS, UD); (**b**) to the CNT distribution (SSSS, $V_{CNT}^{*}=0.17$).

**Figure 5 polymers-14-02664-f005:**
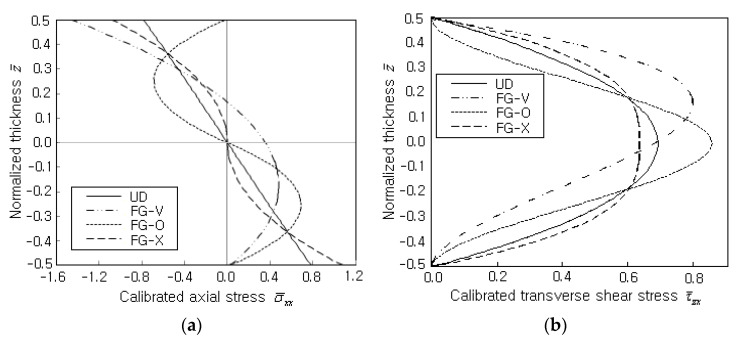
Thickness-wise stress distributions (SSSS, $a/d=50,V_{CNT}^{*}=0.17$): (**a**) calibrated normal stress ${\bar{\sigma }}_{xx}=\sigma_{xx}d^{2}/\left(q_{0}a^{2}\right)$; (**b**) calibrated transverse shear stress ${\bar{\tau }}_{zx}=\tau_{zx}d^{2}/\left(q_{0}a^{2}\right)$.

**Figure 6 polymers-14-02664-f006:**
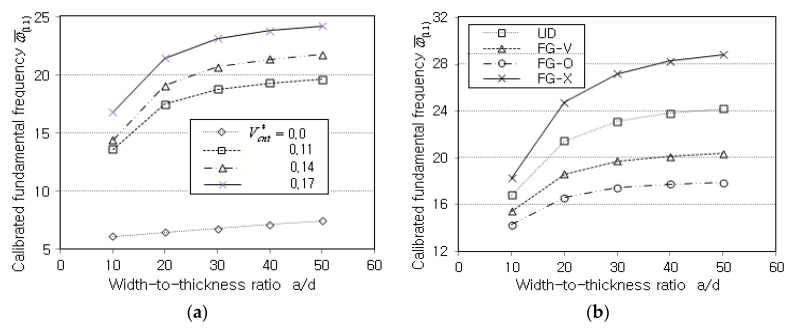
Variation of natural frequencies ${\bar{\omega }}_{\left(1,1\right)}$: (**a**) to the value of $V_{CNT}^{*}$ (UD); (**b**) to the CNT distribution pattern ($V_{CNT}^{*}=0.17$).

**Figure 7 polymers-14-02664-f007:**
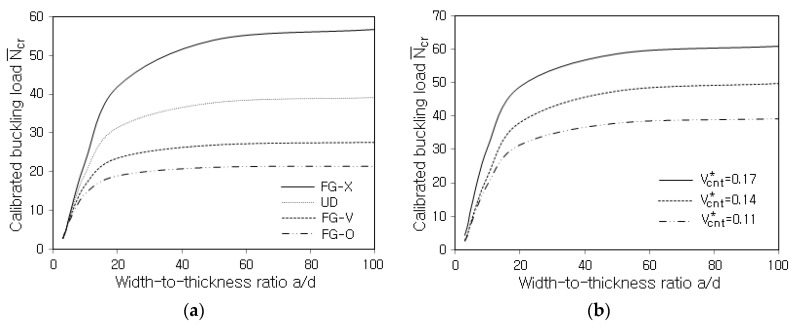
Variation of ${\bar{N}}_{cr}$ to the width-thickness ratio a/d: (**a**) for various CNT distribution patterns; (**b**) for various values of $V_{CNT}^{*}$ (UD).

**Figure 8 polymers-14-02664-f008:**
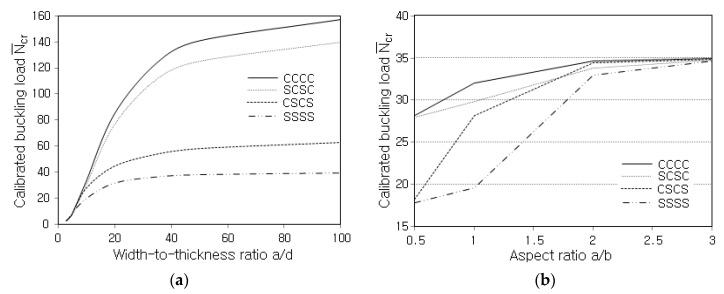
Effect of boundary condition on ${\bar{N}}_{cr}$ of UD: (**a**) for various width-thickness ratios a/d; (**b**) for various aspect ratios a/b (a/d=10).

**Table 1 polymers-14-02664-t001:** The CNT efficiency parameters for three different values of $V_{CNT}^{*}$.

$V_{CNT}^{*}$	$\eta_{1}$	$\eta_{2}$	$\eta_{3}$
0.11	0.149	0.934	0.934
0.14	0.150	0.941	0.941
0.17	0.149	1.381	1.381

**Table 2 polymers-14-02664-t002:** Calibrated central deflection $u_{z}^{c}/d$ with respect to $V_{CNT}^{*}$, a/d, and the CNT distribution.

$V_{CNT}^{*}$	CNTRC	Width-to-Thickness Ratio a/d					
		10		20		50	
		Present	Ref. [20]	Present	Ref. [20]	Present	Ref. [20]
0.11	UD	3.768 × 10^−3^	3.739 × 10^−3^	3.639 × 10^−2^	3.628 × 10^−2^	1.155	1.155
	FG-V	4.481 × 10^−3^	4.466 × 10^−3^	4.905 × 10^−2^	4.879 × 10^−2^	1.658	1.653
	FG-O	5.223 × 10^−3^	5.230 × 10^−3^	6.192 × 10^−2^	6.155 × 10^−2^	2.153	2.157
	FG-X	3.209 × 10^−3^	3.177 × 10^−3^	2.735 × 10^−2^	2.701 × 10^−2^	0.794	0.790
0.14	UD	3.329 × 10^−3^	3.306 × 10^−3^	3.017 × 10^−2^	3.001 × 10^−2^	0.935	0.918
	FG-V	3.906 × 10^−3^	3.894 × 10^−3^	4.068 × 10^−2^	4.025 × 10^−2^	1.356	1.326
	FG-O	4.521 × 10^−3^	4.525 × 10^−3^	5.146 × 10^−2^	5.070 × 10^−2^	1.771	1.738
	FG-X	2.86 × 10^−3^	2.844 × 10^−3^	2.285 × 10^−2^	2.256 × 10^−2^	0.639	0.627
0.17	UD	2.413 × 10^−3^	2.394 × 10^−3^	2.354 × 10^−2^	2.348 × 10^−2^	0.750	0.752
	FG-V	2.875 × 10^−3^	2.864 × 10^−3^	3.185 × 10^−2^	3.174 × 10^−2^	1.080	1.082
	FG-O	3.37 × 10^−3^	3.378 × 10^−3^	4.038 × 10^−2^	4.020 × 10^−2^	1.407	1.416
	FG-X	2.032 × 10^−3^	2.012 × 10^−3^	1.756 × 10^−2^	1.737 × 10^−2^	0.513	0.513

**Table 3 polymers-14-02664-t003:** Calibrated natural frequencies $\bar{\omega }=\omega \left(a^{2}/d\right)\sqrt{\rho^{m}/E^{m}}$ with respect to $V_{CNT}^{*}$ and the CNT distribution.

$V_{CNT}^{*}$	Mode (*m*,*n*)	UD		FG-V		FG-O		FG-X	
		Present	Ref. [20]	Present	Ref. [20]	Present	Ref. [20]	Present	Ref. [20]
0.11	(1,1)	19.265	19.223	16.323	16.252	14.427	14.302	22.923	22.984
	(1,2)	23.859	23.408	21.717	21.142	20.058	19.373	26.847	26.784
	(1,3)	36.131	34.669	34.715	33.350	33.027	31.615	38.406	37.591
	(1,4)	56.365	54.043	54.075	53.430	52.990	51.370	58.430	56.946
	(2,1)	72.560	70.811	63.844	60.188	54.111	53.035	85.007	83.150
	(2,2)	74.799	72.900	64.256	62.780	57.185	55.823	86.811	84.896
0.14	(1,1)	21.394	21.354	18.053	17.995	15.912	15.801	25.495	25.555
	(1,2)	25.607	25.295	23.102	22.643	21.157	20.563	29.094	29.192
	(1,3)	37.525	36.276	35.839	34.660	33.843	32.509	40.344	39.833
	(1,4)	57.741	55.608	56.444	54.833	53.821	52.184	60.439	59.333
	(2,1)	80.100	78.110	67.859	66.552	59.805	58.748	89.442	87.814
	(2,2)	82.079	80.015	70.429	68.940	62.573	61.277	93.039	91.299
0.17	(1,1)	23.743	23.697	20.074	19.982	17.708	17.544	28.326	28.413
	(1,2)	29.523	28.987	26.914	26.204	24.665	23.783	33.455	33.434
	(1,3)	44.881	43.165	43.269	41.646	40.674	38.855	48.301	47.547
	(1,4)	70.129	67.475	68.922	66.943	65.351	63.179	73.793	72.570
	(2,1)	89.534	87.385	75.563	74.030	66.489	65.154	105.213	102.939
	(2,2)	92.360	90.031	79.191	77.343	70.311	68.579	107.597	105.334

**Table 4 polymers-14-02664-t004:** Calibrated uniaxial and biaxial buckling load ${\bar{N}}_{cr}=N_{cr}a^{2}/E_{m}d^{3}$ with respect to $V_{CNT}^{*}$ and the CNT distribution.

Load	$V_{CNT}^{*}$	Method	UD	FG-O	FG-X
Uniaxial	0.11	Ref. [28]	39.2456	21.4573	56.7512
		Present	39.4211	21.6428	56.9679
	0.14	Ref. [28]	49.2112	26.3671	71.8512
		Present	49.6621	26.9072	72.1240
	0.17	Ref. [28]	59.5136	31.2315	87.8012
		Present	60.8003	33.1455	88.3366
Biaxial	0.11	Ref. [28]	11.1118	7.4945	14.2417
		Present	11.2592	7.6119	14.4603
	0.14	Ref. [28]	13.1545	8.3545	16.2198
		Present	13.4774	8.8108	17.6572
	0.17	Ref. [28]	17.2852	11.2654	22.8545
		Present	17.5464	11.7204	22.9129

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
