# Peer review of "Investigation of Mechanical Behaviors of Functionally Graded CNT-Reinforced Composite Plates"

_polymers, 2022, doi:10.3390/polym14132664_

Round 1

Reviewer 1 Report

The paper investigates the bending, free-vibration and buckling behaviors of FG-CNTRC plate by simulation method. In the review, I think this paper should make some revisions.

1. The paper has wrong format, there are blue color words and black words. But I did not know this different colors have meaning for this paper.

2. Used last year's template. Everywhere should be "Polymers 2022, 15". 

3. I think the distribution for the CNT is also an important effect for the reinforcement composites. The author should give some explanation.

4. The author said they investigated the mechanical property, but there are basic mechanical testing for example tensile testing. Why?

5. The microstructural evolution are also important, the author should make some investigations.

Author Response

Please refer to the response to reviewers' comments (1).

Reviewer 2 Report

In the present manuscript, free vibration and buckling analysis of of functionally graded carbon nanotube-reinforced composite plate have been discussed. For achieve this, the displacement is decomposed into the in-field functions and the assumed thickness-wise monomial. The former is defined on the plate midsurface and approximated by the two-dimensional meshfree natural element method. Then some parametric results have been supplied.

From my point of view, this topic is interesting and timely. It is also matched with the journal scope. Hence, I recommend its publication after a revision based on the following comments, 

1     1-In the linguistic point of view, some spelling and grammatical errors are detected that should be corrected. Thus, the English of this work needs some polish.

2-     Introduction is scattered and does not provide the necessary background information. This section should be enriched by giving more details about FGM/CNT or graphene nanoplatelet reinforced and applications via below citations-in remark-3.

1  3-The current references are appropriate. However, the reference list is too brief. Consequently, some new references about the research subject as higher-order plate theories and differnt CMT reinforced models should be added in the reference list and cited properly in the text such. For instance,

     The Influence of the Ceramic Nanoparticles on the Thermoplastic Polymers Matrix: Their Structural, Optical, and Conductive Properties.  Polymers 2021, 13(16), 2773; https://doi.org/10.3390/polym13162773 - 18 Aug 2021.

 Analysis of porous micro-plates reinforced with FG-GNPs based on Reddy plate theory.Composite Structures 247, 112391. https://doi.org/10.1016/j.compstruct.2020.112391

      Free vibration and buckling analyses of CNT reinforced laminated non-rectangular plates by discrete singular convolution method. Engineering with Computers  38, 489–521 (2022)

    Dynamic Characteristics of Woven Flax/Epoxy Laminated Composite Plate. Polymers 2021, 13(2), 209; https://doi.org/10.3390/polym13020209

     4-     What is the main objective behind the current study? It is beneficial for the readers to add more ‎explanations about the novel contribution of this method from theoretical/experimental viewpoints.‎

     5- How author choose Vcnt ratio.

Author Response

Please refer to the response to reviewers' comments (2).

Round 2

Reviewer 1 Report

All comments have modified and replied. The paper could be accepted as this revised form.